# A Simple Empirical Band-Ratio Algorithm to Assess Suspended Particulate Matter from Remote Sensing over Coastal and Inland Waters of Vietnam: Application to the VNREDSat-1/NAOMI Sensor

Dat Dinh Ngoc [1,2,3,*], Hubert Loisel [1,2,3], Vincent Vantrepotte [1,2,3], Huy Chu Xuan [1],
Ngoc Nguyen Minh [1], Charles Verpoorter [2], Xavier Meriaux [2], Hanh Pham Thi Minh [4],
Huong Le Thi [4], Hai Le Vu Hong [5] and Thao Nguyen Van [6]

[1] Vietnam Academy of Science and Technology, Space Technology Institute, 18 Hoang Quoc Viet, Cau Giay, Hanoi 100000, Vietnam; hubert.loisel@univ-littoral.fr (H.L.); vincent.vantrepotte@univ-littoral.fr (V.V.); cxhuy@sti.vast.vn (H.C.X.); nmngoc@sti.vast.vn (N.N.M.)

[2] Laboratoire d'Océanologie et de Géosciences, Univ. Littoral Côte d'Opale, Univ. Lille, CNRS, UMR 8187, LOG, F 59000 Lille, France; charles.verpoorter@univ-littoral.fr (C.V.); xavier.meriaux@univ-littoral.fr (X.M.)

[3] LOTUS, Vietnam Academy of Science and Technology, University of Science and Technology of Hanoi, 18 Hoang Quoc Viet, Cau Giay, Hanoi 100000, Vietnam

[4] Viet Nam Academy of Science and Technology, Institute of Mechanics, 264 Doi Can, Ba Dinh, Hanoi 100000, Vietnam; hanhcmesrc@gmail.com (H.P.T.M.); lthuong211@gmail.com (H.L.T.)

[5] Institute of Techniques for Special Engineering, Le Quy Don University, 236 Hoang Quoc Viet, Bac Tu Liem, Hanoi 100000, Vietnam; lhvh108@gmail.com

[6] Vietnam Academy of Science and Technology, Institute of Marine Environment and Resources, 246 Da Nang, Ngo Quyen, Haiphong 180000, Vietnam; thaonv@imer.vast.vn

* Correspondence: dndat@sti.vast.vn; Tel.: +84-915-089-159

**Abstract:** VNREDSat-1 is the first Vietnamese satellite enabling the survey of environmental parameters, such as vegetation and water coverages or surface water quality at medium spatial resolution (from 2.5 to 10 m depending on the considered channel). The New AstroSat Optical Modular Instrument (NAOMI) sensor on board VNREDSat-1 has the required spectral bands to assess the suspended particulate matter (*SPM*) concentration. Because recent studies have shown that the remote sensing reflectance, $R_{rs}(\lambda)$, at the blue (450–520 nm), green (530–600 nm), and red (620–690 nm) spectral bands can be assessed using NAOMI with good accuracy, the present study is dedicated to the development and validation of an algorithm (hereafter referred to as V1SPM) to assess *SPM* from $R_{rs}(\lambda)$ over inland and coastal waters of Vietnam. For that purpose, an in-situ data set of hyper-spectral $R_{rs}(\lambda)$ and *SPM* (from 0.47 to 240.14 g·m$^{-3}$) measurements collected at 205 coastal and inland stations has been gathered. Among the different approaches, including four historical algorithms, the polynomial algorithms involving the red-to-green reflectance ratio presents the best performance on the validation data set (mean absolute percent difference (*MAPD*) of 18.7%). Compared to the use of a single spectral band, the band ratio reduces the scatter around the polynomial fit, as well as the impact of imperfect atmospheric corrections. Due to the lack of matchup data points with VNREDSat-1, the full VNREDSat-1 processing chain (atmospheric correction (RED-NIR) and V1SPM), aiming at estimating *SPM* from the top-of-atmosphere signal, was applied to the Landsat-8/OLI match-up data points with relatively low to moderate *SPM* concentration (3.33–15.25 g·m$^{-3}$), yielding a *MAPD* of 15.8%. An illustration of the use of this VNREDSat-1 processing chain during a flooding event occurring in Vietnam is provided.

**Keywords:** VNREDSat-1/NAOMI; Landsat-8/OLI; suspended particulate matter; algorithm

---

## 1. Introduction

Climate change, sea level rise, and human activities impact Vietnam coastal regions, which are spread over 3200 km of coastline [1–3]. Climate change and sea level rise will impact Vietnam coastal areas in the near future. Natural hazards (sea water intrusion, flooding, typhons) and human activities, such as aquaculture, sand mining, dam construction, and urban development, represent current threats. For instance, human activities may bring pollutants into the waters [4], reduce the sediment load, especially in Red River [5] and Mekong River deltas [6,7], or increase coastal erosion through the cutting of mangrove for aquaculture activities [8]. Integrating a risk assessment is essential for the development of a suitable strategy for the protection of Vietnamese coastal areas [2]. In that context, spatial remote sensing observation, from which specific spatio-temporal patterns may be identified, is one of the key elements to consider in synergy with in situ monitoring and numerical modeling [9,10].

Ocean color radiometry (OCR) using satellite remote sensing has recently been used for monitoring water surface biogeochemical parameters of Vietnam coastal waters, such as chlorophyll-a [9], Chl-a, and suspended particulate matter (*SPM*) [6,11]. In contrast to offshore waters, where inherent optical properties (IOPs) are mainly driven by phytoplankton and its associated material (the Case-1 waters), coastal and inland waters are optically more complex due to the diversity of the suspended and dissolved organic and inorganic substances, as well as the occurrence of multiple coupled physical/biological/chemical/geological processes covering various temporal scales [12,13]. In contrast to open ocean waters, coastal and inland surface waters are generally characterized by large spatial heterogeneity in terms of bio-optical properties, requiring medium (about 250 m) to high (about 10 m) spatial resolution sensors. *SPM*, which includes both organic and mineral suspended particles, is one of the OCR parameters that has attracted the most attention, mainly due to its implication in many different processes, such as sediment dynamics, coastal erosion processes, and water quality monitoring. Different families of algorithms have been developed to assess *SPM* using remote sensing reflectance, $R_{rs}(\lambda)$, which is the radiometric parameter estimated after atmospheric correction. Among these different approaches, one may distinguish purely empirical algorithms, based on a single band, multiple bands, or band ratios [14–18] and semi analytical algorithms [19–21] generally involving IOPs.

The first Vietnamese optical satellite, VNREDSat-1, which was launched on 7 May 2013, carries the New AstroSat Optical Modular Instrument (NAOMI), which has four multi-spectral bands (Table 1) at 10 m of spatial resolution and 1 panchromatic band at 2.5 m [22,23]. The water pixel extraction (referred to as WiPE) and atmospheric correction (referred to as RED-NIR) algorithms for VNREDSat-1 have recently been developed [24], allowing to estimate $R_{rs}(\lambda)$ at each NAOMI spectral band. Bio-optical algorithms can now be used to assess biogeochemical properties of surface waters from the NAOMI $R_{rs}(\lambda)$ spectra. For the different reasons mentioned above, the objective of the current paper is to develop an algorithm to estimate *SPM* from NAOMI over Vietnam coastal and inland waters (hereafter referred to as V1SPM). For that purpose, we first present the in-situ data set, which has been used for the development and validation of the algorithm. The performance of V1SPM is then evaluated and compared to existing algorithms on the in-situ validation data set. Unfortunately, no in situ *SPM* measurements were collected during the VNREDSat overpass; therefore, this new *SPM* algorithm was applied to Landsat-8/OLI observations performed nearly simultaneously to some water sampling in Vietnam and Cambodia coastal and inland waters. For that purpose, the OLI scenes have been processed using the NAOMI algorithms, that is the NAOMI RED-NIR atmospheric correction [24] and V1SPM algorithms, to first assess the coherence of NAOMI algorithms to assess *SPM*. The pertinence of this match-up exercise procedure, developed from OLI observations but processed using the NAOMI algorithms, has already been partly discussed in [24] for the validation of the RED-NIR atmospheric correction algorithm. Inter-comparison of the NAOMI *SPM* products is then performed at the Camau Peninsula (South of Vietnam), observed nearly simultaneously by NAOMI, processed using the Red-NIR and V1SPM algorithms and OLI, and processed using the last version of ACOLITE [25].

**Table 1.** VNREDSat-1 spectral bands, spatial resolution, gain, and bias values. The two later coefficients are used to calculate the top of atmosphere radiometric signal from the measured digital number using Equation (4) in [24].

| Band | Wavelength (nm) | Resolution (m) | Gain | Bias |
|---|---|---|---|---|
| B1–Blue | 450–520 | 10 | 1.638 | 0 |
| B2–Green | 530–600 | 10 | 1.621 | 0 |
| B3–Red | 620–690 | 10 | 1.848 | 0 |
| B4–Near Infrared | 760–890 | 10 | 2.511 | 0 |
| B5–Panchromatic | 450–740 | 2.5 | 1.950 | 0 |

## 2. Material and Methods

### 2.1. In Situ Data Sets

In situ data (N = 205) of remote sensing reflectance ($R_{rs}$) and *SPM* were collected in Vietnam coastal waters (Figure 1) from 2011 to 2015 during 7 field surveys, performed during different time periods of the year, as already described in [9]. This in situ data set gathered data collected in 3 different locations. About 60% of the data set (N = 131, 1.05–147.69 g·m$^{-3}$, 17.62 ± 25.96) was composed of data collected in the North Vietnam in water masses under the influence of the Red River, as well as, in typical case 1, waters for which the inherent optical properties are driven by phytoplankton and its associated material. The second data set, which represents 22% of the whole data set, (N = 46, 1.01–240.14 g·m$^{-3}$, 22.68 ± 48.98) was composed of measurements performed in South Vietnam in water masses impacted by the Mekong River outputs [5,20]. In these highly turbid waters, measured *SPM* concentration reached up 240.14 g·m$^{-3}$. In addition, samples were collected in the Nhatrang coastal waters, where relatively low *SPM* values (N = 28, 0.47–7.7 g·m$^{-3}$, 1.85 ± 1.74) were sampled [4]. The protocol used to calculate the $R_{rs}(\lambda)$ spectra from radiometric measurements is fully described in [9]. Due to the large fluctuation of salinity in the data set (28.73 ± 2.26), which was driven by impact of fresh water inputs at some stations and which may bias the *SPM* measurements [26,27], great care was taken to properly remove salt from the filters, which were gently washed with deionized water. Three replicates were made at each station, and only stations with standard deviation lower than 20% were kept, allowing to reduce the impact of measurement uncertainties.

The latter in situ data set was divided into two subsets, one for developing the *SPM* algorithm over coastal waters of Vietnam (D-DS, N = 143) and the second for the validation exercise (V-DS, N = 62). D-DS and V-DS comprised 70 and 30% of the initial in situ data set (Table 2), respectively. *SPM* distribution for the two latter data sets was similar (Figure 2), ensuring an unbiased evaluation of new the *SPM* algorithm for VNREDSat-1/NAOMI from V-DS. Hyperspectral $R_{rs}$ spectra were measured in situ using Trios radiometers (see the methodology in Loisel et al., 2017). Multi-spectral $R_{rs}$ corresponding to the bands of VNREDSat-1/NAOMI were generated from these hyperspectral data for each in situ sample, as follows:

$$R_{rs}(\lambda_i) = \frac{\int_{j=p}^{j=q} W_j R_{rs}(\lambda_j)}{\int_{j=p}^{j=q} W_j} \tag{1}$$

where $R_{rs}(\lambda_i)$ is the multispectral remote sensing reflectance for the VNREDSat-1 band *i* (*i* = 1; 2; 3; 4) ranging over the spectral domain [*p*, *q*] (Table 1) and $W_j$ is the hyperspectral weight applied to the hyperspectral remote sensing reflectance $R_{rs}(\lambda_j)$ spanning over each VNREDSat-1 band provided by the satellite manufacturer (Figure 3). Sensor specific $R_{rs}$ values were also computed (using the V-DS validation dataset) for OLI and MODIS in order to enable the evaluation of other sensor specific models for estimating *SPM* concentration in the coastal waters of Vietnam using the respective spectral response functions documented for these two latter sensors (OLI SRF: https://landsat.gsfc.nasa.gov/

wp-content/uploads/2013/06/Ball_BA_RSR.v1.1-1.xlsx; MODIS SRF: https://oceancolor.gsfc.nasa.gov/docs/rsr/HMODISA_RSRs.txt).

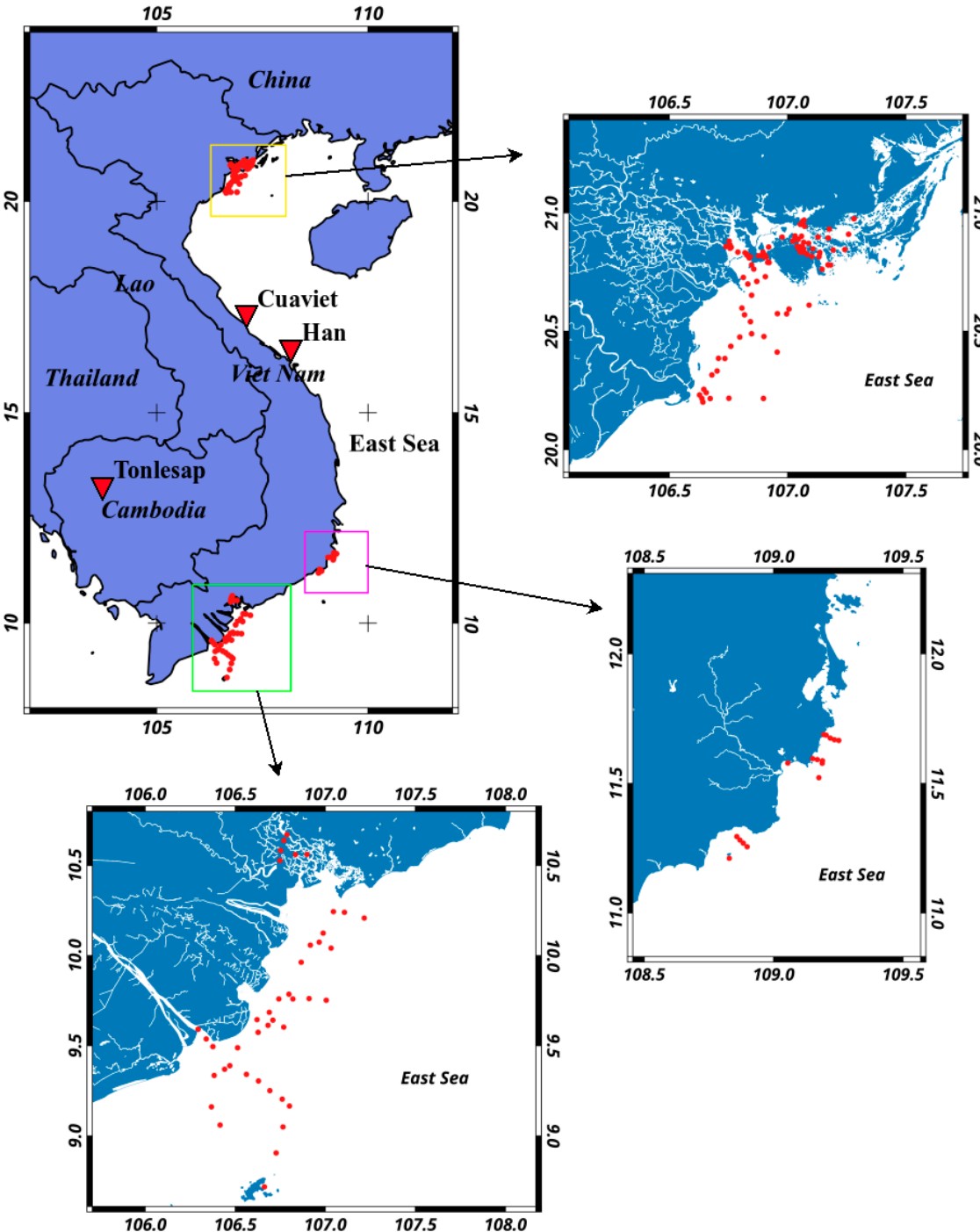

**Figure 1.** Location of the in situ data collected over Vietnamese coastal waters (Yellow: Red River influenced waters, Green: Mekong influenced waters, Pink: Nhatrang coastal waters).

**Table 2.** Suspended particulate matter (*SPM*) (g·m$^{-3}$) statistics for the development (D-DS) and validation (V-DS) of in situ subsets collected over the Vietnamese coastal waters. N indicates the number of samples for each data set.

| Data Set | N | Min, Max | Geometric Mean | Standard Deviation |
|---|---|---|---|---|
| Development (D-DS) | 143 | 0.47, 188.12 | 5.44 | 27.51 |
| Validation (V-DS) | 62 | 0.52, 240.14 | 6.84 | 42.77 |
| Overall | 205 | 0.47, 240.14 | 5.68 | 32.85 |

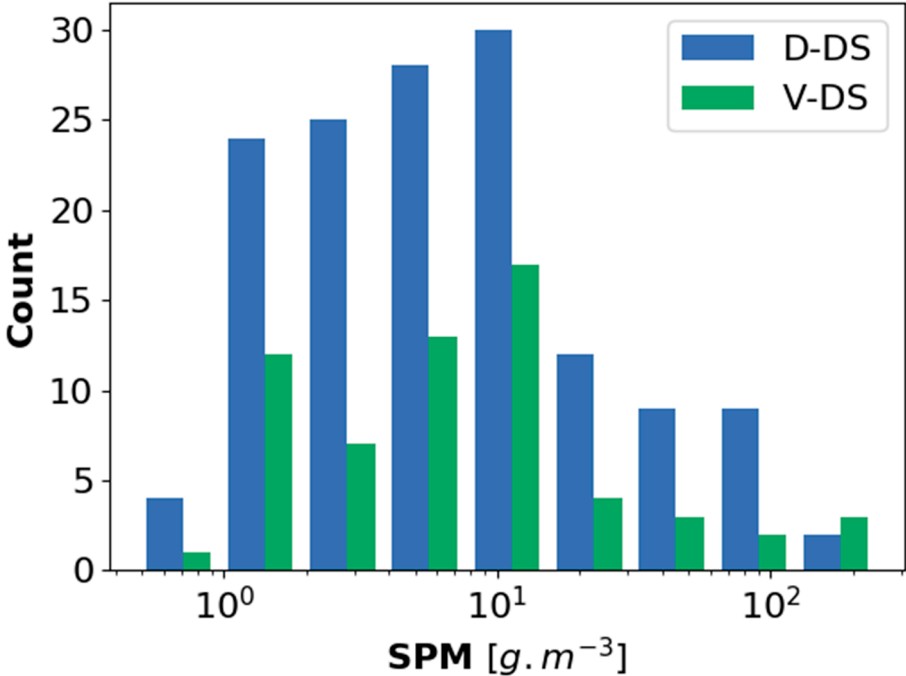

**Figure 2.** In situ *SPM* frequency distribution histograms for the development (D-DS, N = 143) and validation (V-DS, N = 62) subsets.

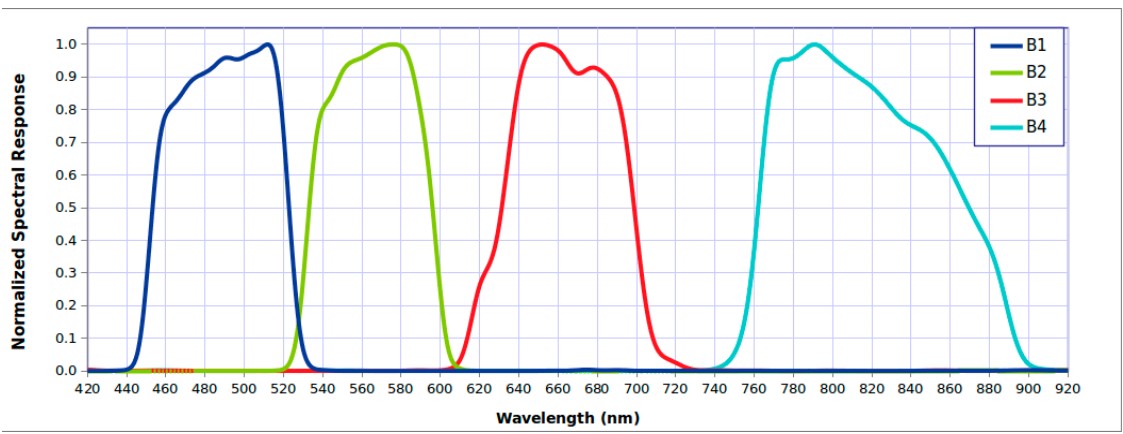

**Figure 3.** Spectral response functions for each VNREDSat-1 band.

In addition to the two previous data sets, an independent *SPM* in situ matchup data set (M-DS, N = 51) for Landsat-8/OLI (Table 3) was collected for validating the new *SPM* algorithm over Vietnamese coastal waters and inland waters. This dataset covered three study areas (Tonlesap, Cambodia, Cuaviet Estuary, Han Estuary), showing relatively low to moderate *SPM* concentration

(3.33–15.25 g·m$^{-3}$, Table 3). The time difference between the in situ and satellite overpass was always lower than 4 h.

**Table 3.** *SPM* (g·m$^{-3}$) statistics for the Landsat-8/OLI matchup dataset (M-DS). N indicates the number of samples for each data set.

| Data Set | N | Min, Max | Geometric Mean | Standard Deviation |
|---|---|---|---|---|
| Tonlesap, Cambodia (13 October 2018) | 18 | 3.33, 15.25 | 5.85 | 3.00 |
| Cuaviet estuary, Vietnam (25 April 2019) | 20 | 8.30, 14.90 | 10.20 | 1.60 |
| Han estuary, Vietnam (4 May 2019) | 13 | 6.84, 14.58 | 10.54 | 2.37 |
| Overall | 51 | 3.33, 15.25 | 8.45 | 3.07 |

## 2.2. Satellite Match-Up Data Set

Landsat-8/OLI image Level-1 data were downloaded from the Glovis portal (https://glovis.usgs.gov) of the US Geological Survey (USGS). Water pixels were identified using the WiPE masking procedure [28]. Since there was no direct matchup data for VNREDSat-1/NAOMI, an indirect evaluation of the VNREDSat-1/NAOMI derived from *SPM* was performed using Landsat-8/OLI images processed using the RED-NIR NAOMI atmospheric correction [24] and V1SPM algorithm.

## 2.3. Evaluated Historical SPM Inversion Algorithms

Initially, four existing algorithms were considered. The selected *SPM* inversion algorithms differed in terms of the sensor specific development assumption and varied regarding the *SPM* range, through which each algorithm was developed, alongside the development data set used, which may be different in terms of inherent optical properties driven by the concentration, chemical nature (mineral vs. organic), and particulate size distribution of the optically significant matter.

Doxaran et al. (2005) [29] developed a *SPM* inversion algorithm for an application on SPOT5 images, which are relatively similar, in terms of spatial and spectral resolution, to those provided by VNREDSat-1/NAOMI. This model (hereafter referred to as Doxaran-SPOT5), based on the NIR/green band ratio, is expressed as follows:

$$SPM = 26.083 \times e^{(0.336X)} \tag{2}$$

where $X = [R_{rs}(865)/R_{rs}(555)]$.

Nechad et al. developed a *SPM* inversion algorithm [19] that can be applied on diverse sensors, including Landsat-8/OLI and Sentinel-2/MSI [30,31]. This model is based on the marine reflectance in the red part of the spectrum. Applications on Landsat-8/OLI Nechad's model (hereafter referred to as Nechad-OLI) is expressed as follows:

$$SPM = \frac{384.11\rho_w(655)}{1 - \frac{\rho_w(655)}{0.1747}} + 1.44 \tag{3}$$

where $\rho_w(655) = \pi R_{rs}(655)$.

A general semi analytical *SPM* algorithm documented by Han et al. [20] was designed for estimating surface *SPM* on a global scale from OCR based on a large in situ data set, covering 3 orders of magnitudes of *SPM* concentration, which has been optimized for a variety of OCR sensors. The model proposed by Han et al. [20] for Landsat-8/OLI applications (hereafter referred to as Han-OLI) is expressed as follows:

$$SPM = \frac{W_L \cdot SPM_L + W_H \cdot SPM_H}{W_L + W_H} \tag{4}$$

where

$$W_L = \begin{cases} 1, & if\ R_{rs}(655) \leq 0.03 \\ 0, & if\ R_{rs}(655) \geq 0.045 \\ log_{10}(0.045) - log_{10}[R_{rs}(655)], & otherwise \end{cases}$$

$$W_H = \begin{cases} 0, & if\ R_{rs}(655) \leq 0.03 \\ 1, & if\ R_{rs}(655) \geq 0.045 \\ log_{10}[R_{rs}(655)] - log_{10}(0.03), & otherwise \end{cases}$$

Finally, the multiband *SPM* algorithm developed by Siswanto for MODIS regional applications [16] was also selected. It is expressed as follows:

$$log_{10}(SPM) = 0.649 + 25.623X_1 - 0.646X_2 \tag{5}$$

where $X_1 = R_{rs}(555) + R_{rs}(659)$ and $X_2 = R_{rs}(490)/R_{rs}(555)$.

The performance of the latter four historical models, documented for different medium and low spatial resolution sensors, was assessed and compared to that of the *SPM* VNREDSat-1/NAOMI specific model further developed in this work over the Vietnamese coastal waters.

## 2.4. Statistical Indicators

These *SPM* algorithms were evaluated considering diverse statistical indicators, including the root mean square deviation in log space ($RMSD_{log}$) and the mean absolute percent difference (*MAPD*). The $RMSD_{log}$ and *MAPD* were computed as follows:

$$RMSD_{log} = \sqrt{\frac{\sum_{k=1}^{N}(log(SPM^k_{estimated}) - log(SPM^k_{observed}))^2}{N}} \tag{6}$$

$$MAPD = 100 \cdot \frac{1}{N}\sum_{k=1}^{N}\left|\frac{SPM^k_{estimated} - SPM^k_{observed}}{SPM^k_{observed}}\right| \tag{7}$$

where $SPM^k_{observed}$ and $SPM^k_{estimated}$ represent the measurement and estimated *SPM* values for a defined station *k*.

## 3. Results and Discussion

### 3.1. Development of a SPM Inversion Algorithm for VNREDSat-1/NAOMI

Historical *SPM* algorithms were developed for past ocean color sensors and were not specifically optimized for Vietnamese coastal waters. A new *SPM* inversion algorithm (referred to as V1SPM) was therefore developed to improve the accuracy of *SPM* estimation from VNREDSat-1/NAOMI reflectance maps. *SPM* inversion algorithms were evaluated considering various possible VNREDSat-1/NAOMI single band or band ratio formalisms based on the D-DS data set.

The best single band algorithm found was based on a cubic transform relationship (Figure 4a) between $Log_{10}[R_{rs}(3)]$ and $Log_{10}(SPM)$ ($R^2 = 0.904$, Equation (8)), although a relative high scatter was observed around the mean. This scatter can be expected in coastal environments where the reflectance signal at visible wavelengths is impacted by the illumination condition [32–34] and sediment type [17] (Figure 5a). The use of band ratio formalisms allows such effects to be reduced [35]. The best band ratio found from the D-DS data set corresponds to a cubic transform relationship between $Log_{10}[R_{rs}(3)/R_{rs}(2)]$ and $Log_{10}(SPM)$ (Equation (9), $R^2 = 0.949$, Figure 4b). Globally, a better performance ($R^2 = 0.949$) and a lower scatter was observed for this band ratio algorithm when compared to the single band algorithm.

The improvement of the band ratio-based inversion model was particularly visible for the highest *SPM* loads (>100 g·m$^{-3}$).

$$log_{10}(V1SPM) = 0.281log_{10}{}^3R_{rs}(3) + 2.48log_{10}{}^2R_{rs}(3) + 7.94log_{10}R_{rs}(3) + 9.35 \quad (8)$$

$$log_{10}(V1SPM) = 0.663log_{10}{}^3\left(\frac{R_{rs}(3)}{R_{rs}(2)}\right) + 1.48log_{10}{}^2\left(\frac{R_{rs}(3)}{R_{rs}(2)}\right) + 2.57log_{10}\left(\frac{R_{rs}(3)}{R_{rs}(2)}\right) + 1.59 \quad (9)$$

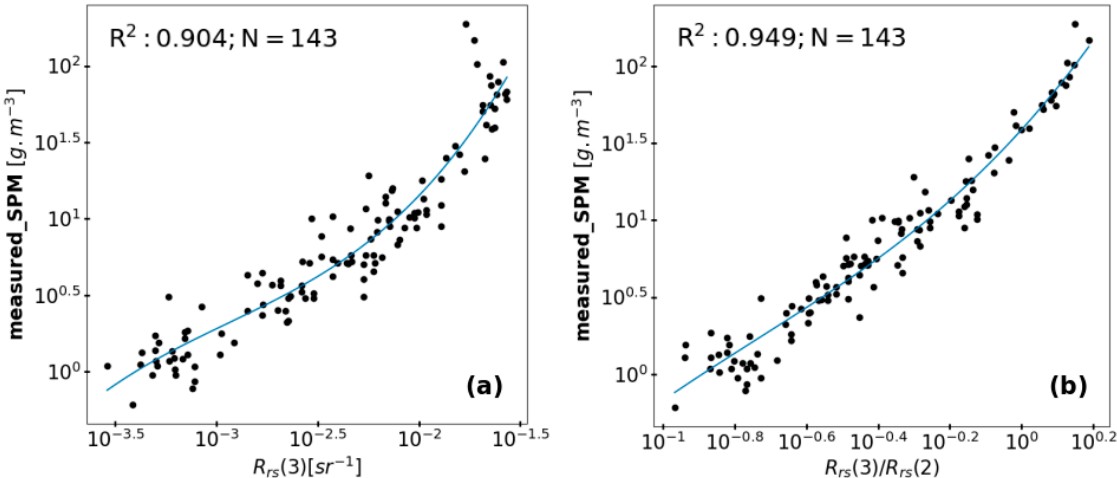

**Figure 4.** (**a**) Cubic transformed relationships between in situ remote sensing reflectance corresponding to the VNREDSat-1/NAOMI red band, $R_{rs}(3)$, in situ *SPM*, and (**b**) between the cubic transformed relationship between the VNREDSat-1/NAOMI Red/Green band ratio, $R_{rs}(3)/R_{rs}(2)$, and in situ *SPM*.

Considering the previous results, the VNREDSat-1/NAOMI regional *SPM* inversion algorithm based on the $R_{rs}(3)/R_{rs}(2)$ band ratio was selected for further computing VNREDSat-1/NAOMI images, validating the *SPM* estimates and comparing the performance of this new formalism with that of historical models.

### 3.2. Inter-Comparison and Validation of SPM Inversion Algorithms for Medium Spatial Sensors over Vietnamese Coastal Waters

Hyperspectral in situ remote sensing reflectance spectra were used to compute the reflectance signal corresponding to the VNREDSat-1/NAOMI (this work), SPOT-5 [29] Landsat-8/OLI [19,20], and MODIS [16] bands using the spectral response function respective to each of the sensors in order to compare the performance of the various sensor specific models proposed for estimating *SPM* in an application over the Vietnamese coastal waters.

Results obtained on the V-DS validation data set are presented in the Figure 6 and Table 4. SPOT 5-Doxaran (*MAPD* = 218%) showed the lowest performance for retrieving *SPM* in the Vietnamese coastal waters, although the latter formulation provided relevant *SPM* retrievals in the most turbid waters (*SPM* > 35 g·m$^{-3}$). This result was expected as the empirical SPOT-5-Doxaran algorithm was developed from 42 in situ *SPM* data points with concentrations ranging from 35 to 2072 mg·L$^{-1}$. Conversely, the four other *SPM* algorithms evaluated on V-DS (namely Nechad-OLI, Siswanto, Han-OLI, and V1SPM-NAOMI) showed relevant prediction accuracy due to their general character and thus due to the fact that they were developed on a larger *SPM* range than the two previous models. *V1SPM* had the highest prediction (*MAPD* = 18%) and measurement accuracy (*RMSD$_{log}$* = 0.188) in comparison with other selected SPM algorithms. Note that the use of the single band formulation (Equation (8)) tended to generate a higher scatter in the SPM retrieval (*MAPD* = 27.47%), underlining the interest of using band ratio formulation for minimizing the impact of atmospheric correction imperfection.

A gain in precision was seen, provided by the V1SPM-NAOMI model when compared to Nechad-OLI, Siswanto, and Han-OLI. Indeed, Nechad-OLI (Figure 6b) estimates were sharply underestimated at *SPM* levels higher than 100 g·m$^{-3}$, while an overestimation was found for SPM loads lower than the latter concentration. Siswanto and Han-OLI (Figure 5c,d, respectively) conversely underestimated *SPM* over the whole range of values in the V-DS dataset. Despite the general better performance for the *V1SPM* algorithm (Figure 6e), a slight underestimation of the *SPM* loads was observed at the highest levels (>100 g·m$^{-3}$), while a low *SPM*, where measurement uncertainties can occur (e.g., salt retention issues) was slightly overestimated.

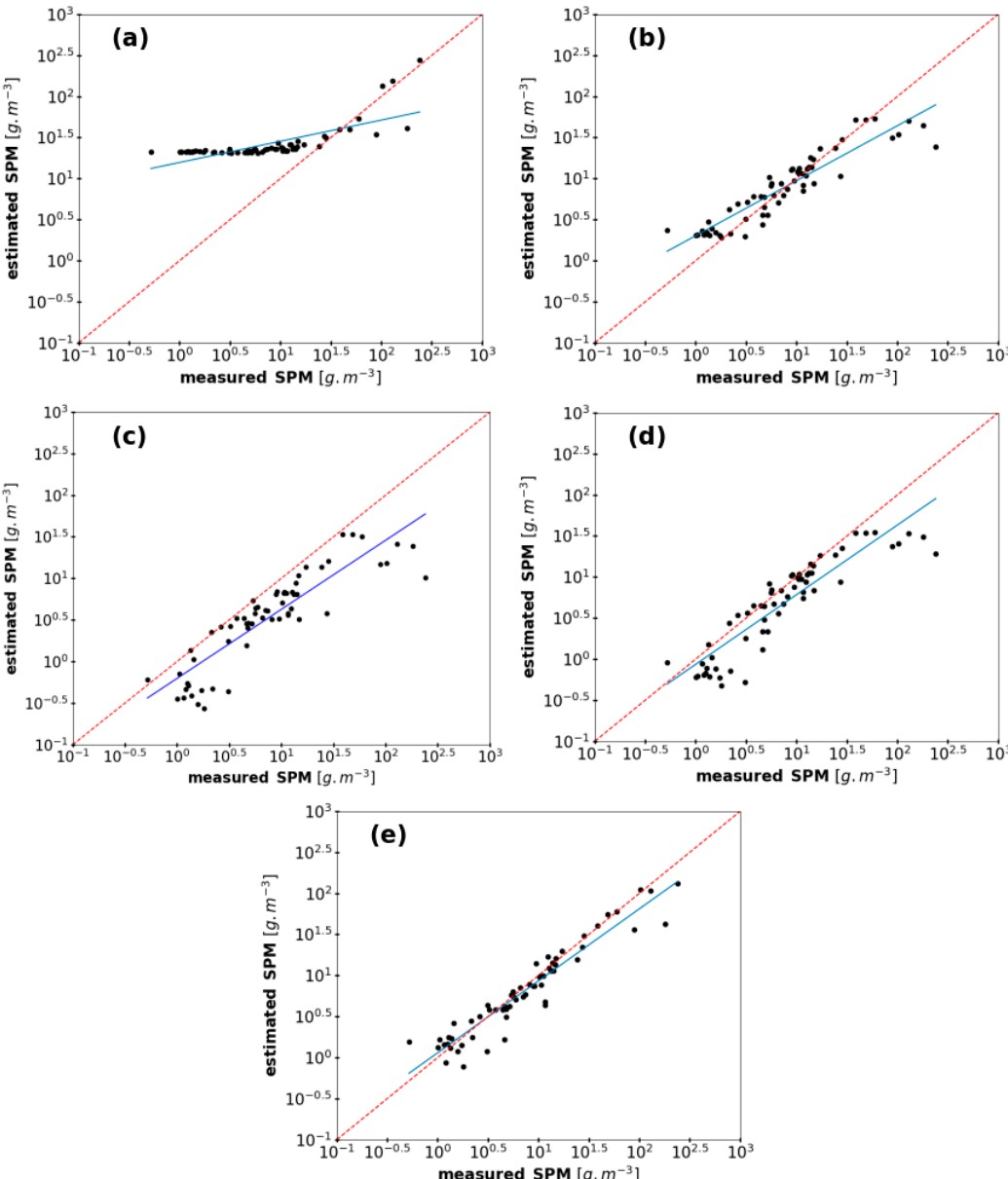

**Figure 5.** Inter-comparison of *SPM* inversion algorithms over Vietnamese coastal waters using the V-DS validation data set (N = 62) Doxaran SPOT 5 (**a**), Nechad algorithm OLI (**b**), Siswanto algorithm (**c**), Han algorithm OLI (**d**), and *V1SPM* algorithm NAOMI (**e**). The dashed red line represents the 1:1 line and the blue line represents the regression line between the measured and estimated values for each model.

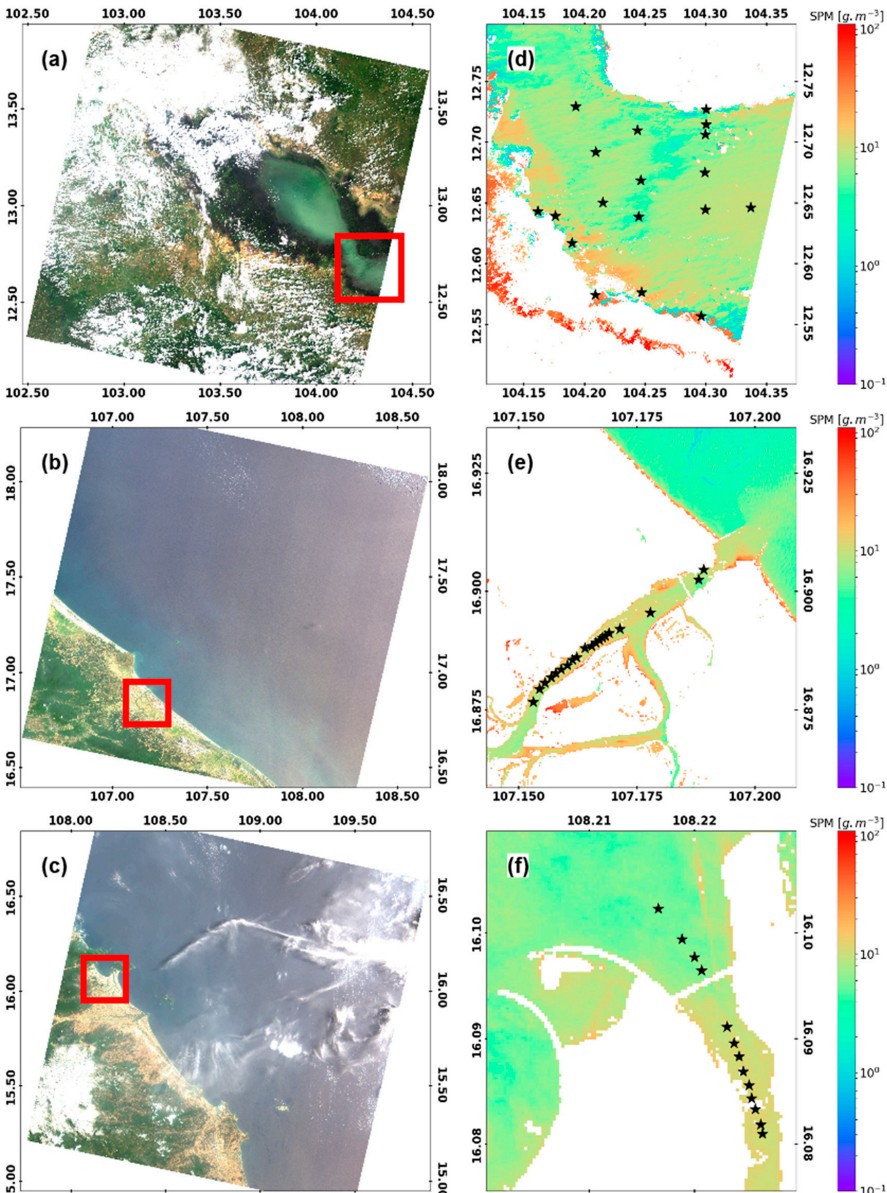

**Figure 6.** RGB Landsat-8/OLI composited images acquired over (**a**) the Tonle Sap lake (Cambodia) on 13 October 2018, (**b**) the Quangtri province (Vietnam) on 25 April 2019, and (**c**) in Danang (Vietnam) on 4 May 2019. Results of the V1SPM algorithm over the sub-areas delimited by the red squares over the (**d**) Tonle Sap lake, (**e**) Cuaviet Estuary, and (**f**) Han Estuary. Black stars provide the location of the different match-up stations.

**Table 4.** Performance statistics of the *SPM* retrieval on the validation data set (V-DS, N = 62) for the different historical and new inversion algorithms.

| Algorithm | *MAPD* (%) | $RMSD_{log}$ | Slope | Bias | $R^2$ |
|-----------|-----------|-------------|-------|------|-------|
| Doxaran | 218 | 0.725 | 0.258 | 1.200 | 0.54 |
| Nechad | 36 | 0.255 | 0.670 | 0.304 | 0.86 |
| Siswanto | 46 | 0.45 | 0.831 | −0.203 | 0.78 |
| Bing Han | 30 | 0.321 | 0.848 | −0.064 | 0.81 |
| *V1SPM* | 18 | 0.188 | 0.877 | 0.058 | 0.91 |

### 3.3. Validation of the Red-NIR/V1SPM Algorithms Using OLI Observations

Due to the lack of *SPM* match-up data points for VNREDSat-1, the *V1SPM* algorithm was applied to the OLI sensor, for which 51 *SPM* match-up data points were available (Figure 6). The considered OLI scenes were processed using the RED-NIR atmospheric correction algorithm [24], using only the spectral bands in common between OLI and NAOMI. This latter configuration allowed the performance of the whole VNREDSat processing chain dedicated to the estimation of *SPM* to be fully addressed. Considering the whole match-up data set built from the 3 different areas, the RED-NIR/V1SPM algorithm was able to estimate *SPM* with excellent accuracy (Table 5 and Figure 7). For instance, the *MAPD* and bias values were 15.79% and 1.43 g·m$^{-3}$, respectively. Except for the Cuaviet Estuary, the performance only slightly differed according to the sub-match-up data set used (Table 5), which mainly reflected the impact of a different distribution of the data rather than a true regional variability. The impact of the regional discrepancies could indeed only be addressed by a higher number of match-up data points. The lowest performance of the algorithm observed at the Cuaviet Estuary (smaller than the Han Estuary) could be explained by the complex hydrological condition of the Estuary, bottom albedo effect, and adjacency effect.

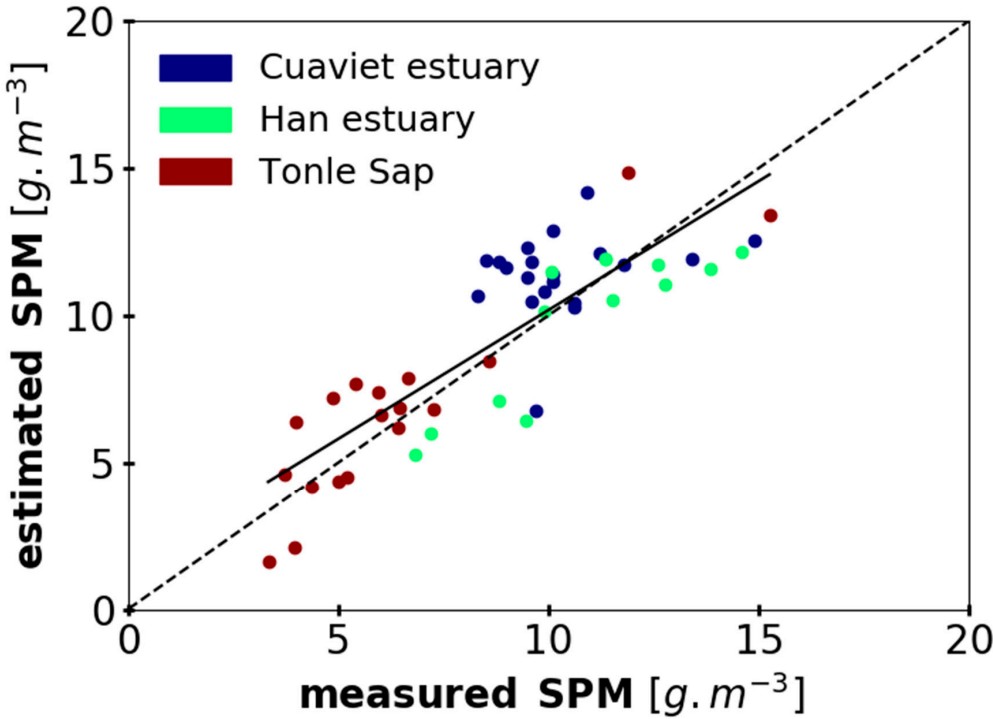

**Figure 7.** Comparison of the measured and inversed *SPM* values from the Landsat-8/OLI match-up data set (N = 51) at the three different locations, as indicated (Tonle Sap, Cuaviet Estuary, and Han Estuary). The solid line represents the best linear regression fit to data and the dashed line represents the 1:1 line.

**Table 5.** Statistical indicators of the performance of the RED-NIR/V1SPM algorithm over the 51 *SPM* match-up data points.

| Data Set | *MAPD* (%) | $RMSD_{log}$ | Slope | Bias | $R^2$ |
|----------|-----------|--------------|-------|------|-------|
| Tonle Sap | 15.87 | 0.13 | 0.979 | 0.515 | 0.793 |
| Cuaviet | 17.33 | 0.09 | 0.243 | 8.987 | 0.068 |
| Han | 13.86 | 0.08 | 0.931 | −0.265 | 0.734 |
| All | 15.79 | 0.10 | 0.875 | 1.431 | 0.703 |

Satellite *SPM* products are now compared over the Camau (South of Vietnam) coastal area, which has been sampled almost simultaneously by Landsat-8/OLI (at 03:14:41.7 UTC time) (Figure 8a) and VNREDSat-1/NAOMI (03:51:14.2 UTC time) (Figure 8b) on 8 January 2015, that is, during the dry season, which generally occurs from November to April. During this period, suspended sediments are generally concentrated along the Eastern Camau peninsula coast, where the sediment dynamics are basically controlled by the Mekong River inputs during summer and resuspension effects during winter [6]. Other images were also available for this inter-comparison exercise but have not been considered due to glitter contamination. The Landsat-8/OLI and VNREDSat-1/NAOMI images were processed using the last version of ACOLITE [25] (Vanhellemont, 2019) and Red-NIR/V1SPM algorithms, respectively. Due to the different spatial resolution of the two considered sensors, the VNREDSat-1/NAOMI pixels were resized from 10 m to 30 m, using the nominal spatial resolution of Landsat-8/OLI. The inter-comparison was performed on a sub-area of the images to limit the impact of white caps (see discussion in [28]). A relatively good agreement between the two *SPM* products could be observed, with more pixels with higher *SPM* values for OLI than NAOMI (Figure 8e).

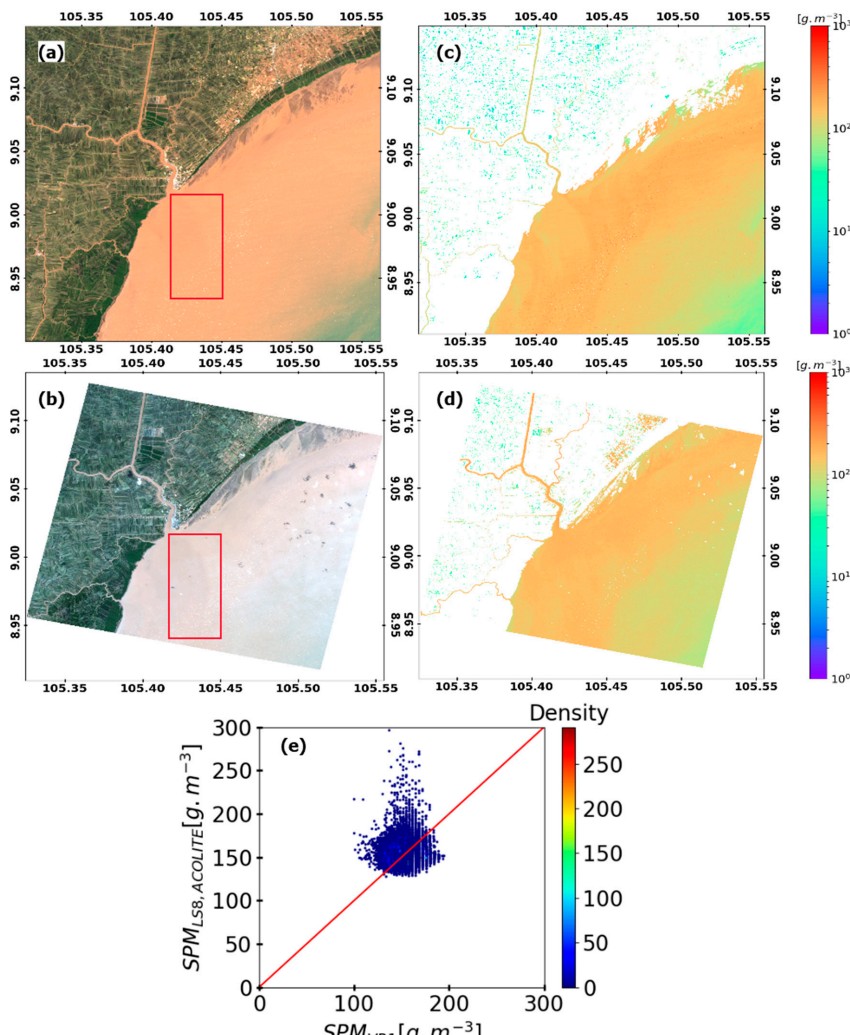

**Figure 8.** Landsat-8/OLI (**a**) and VNREDSat-1/NAOMI (**b**) RGB composited at coastal of Camau province in South of Vietnam on 8 January 2015. The captured time at central scenes are 03:14:41.7 UTC time (Landsat-8) and 03:51:14.2 UTC time (VNREDSat-1). In the panel (**e**), VNREDSat-1/NAOMI *SPM* (**d**) by V1SPM estimation is compared with Landsat-8/OLI *SPM* (**c**) estimation by the Nechad algorithm used in ACOLITE over the red rectangle areas in (**a**) and (**b**). The 1:1 line is drawn in red.

### 3.4. Illustration of the Monitoring of a Flooding Event from VNREDSat-1

An illustration of how the RED-NIR and V1SPM algorithms can be used to monitor environmental issues from VNREDSat-1 is provided in Figure 9. Due to heavy rainfall and water released by the Vucmau hydro power dam, a large flooding event occurred in the Nghean province (Figure 9a) from 22 September 2013 to 1 October 2013. A series of VNREDSat-1 images were acquired during the flooding time period (Figure 9b) and 3 (Figure 9c) and 8 (Figure 9d) days after this event. This very short time series captured by VNREDSat-1 shows the spatial coverage of the flooding, as well as the rapid retreat of waters following this natural hazard event. Information about the concentration of *SPM* is also provided over each flooded area, as well as at the outlet of the Yen River, where the river plum areas sharply decrease in 2 weeks.

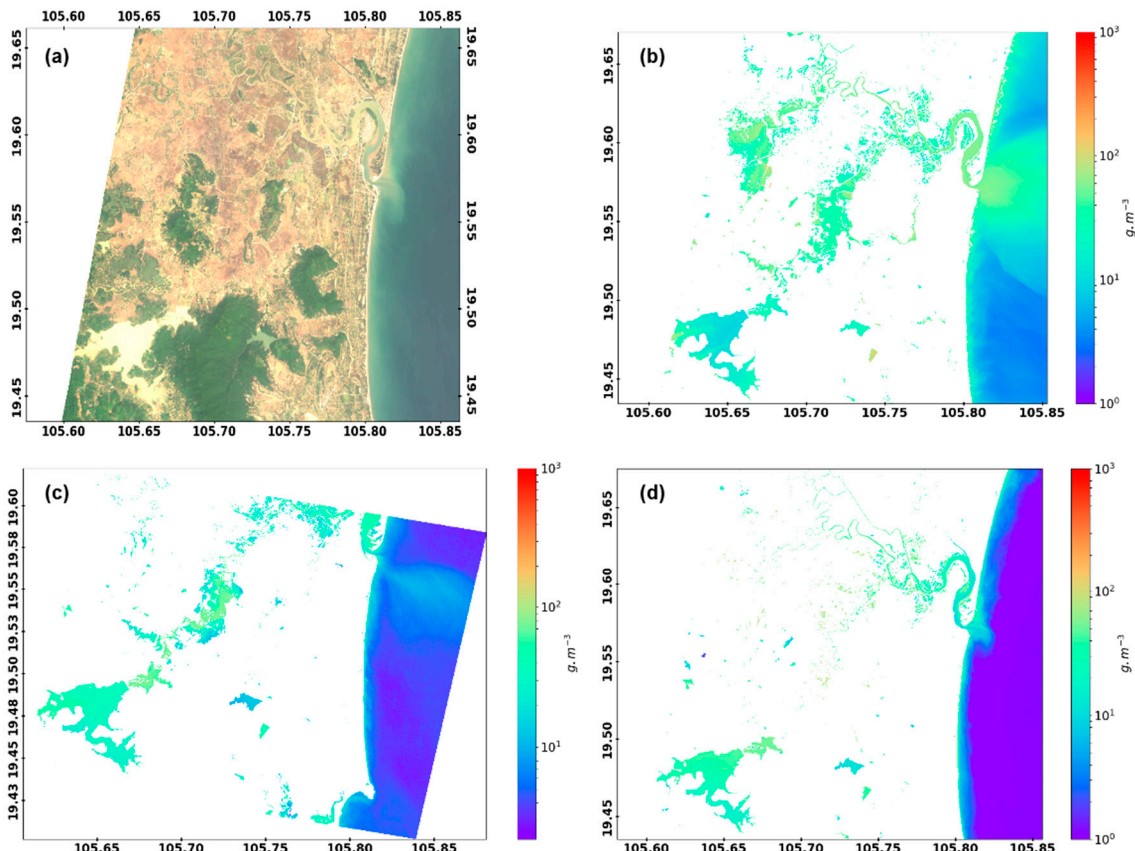

**Figure 9.** (**a**) RGB map over the Nghean province showing the Vucmau reservoir. *SPM* maps processed by the atmospheric correction (RED-NIR) and V1SPM algorithms, acquired on (**b**) 22 September 2013 and (**c**) 4 and (**d**) 8 October 2013.

## 4. Concluding Remarks

The present study was dedicated to the development of an *SPM* algorithm for the VNREDSat-1/NAOMI sensor. The simple empirical formulation proposed here, based on a band-ratio approach, offers, at least for the present data set, the best solution for the *SPM* retrieval at a regional scale. Additionally, this ratio allows to reduce the scatter around the adopted polynomial fit between radiometric measurements and *SPM*. However, further researches should be undertaken to better understand and qualify the origin of this scatter (measurement uncertainties vs. natural variability), as previously done, for instance, with algorithms aiming at estimating Chl-a [36]. At last, we recommend that a proper validation plan for NAOMI should also be undertaken in the very near future.

**Author Contributions:** D.D.N. developed final model and finished the original manuscript. H.L. proposed the idea of the work, gave permission to data at LOG, and did the writing—review and editing. V.V. validated final model and did the writing—review and editing. H.C.X. and N.N.M. worked with the VNREDSat-1/NAOMI database to search the match-ups with Landsat-8/OLI. C.V. and X.M. collected in situ data over Tonlesap. H.P.T.M. and H.L.T. collected in situ data over Cuaviet Estuary and Han Estuary. T.N.V. and H.L.V.H. did VNREDSat-1 quality checking, such as removing cloud scenes and processing satellite images. All authors have read and agreed to the published version of the manuscript.

**Funding:** This study is funded by the VT-UD.02/17-20 project, which belongs to the Vietnam National Space Science and Technology Program, QTRU04.07/18-19 project, under the Space Technology Institute and 2018.73.041 project, under Le Quy Don University.

**Acknowledgments:** The authors would like to thank satellite image distributors, including USGS for Landsat-8/OLI and the center of control and exploitation of small satellites, the Space Technology Institute, and the Vietnam Academy of Science and Technology for distribution of Level-1 data products. The in situ data set used in this study was collected in the frame of the VITEL (CNES, TOSCA), Black Carbon, VolTransMEKONG (TOSCA/CNES), and Vietnam National Space Science and Technology Program projects. The VITEL cruises in 2014 were performed onboard the research vessel ALIS from IRD. The authors are also thankful to the Vietnam National Space Science and Technology Program, LOG/ULCO, University of Science and Technology of Hanoi, the Space Technology Institute, and the Institute of Technology of Technology of Cambodia for funding and facilities for this research. The authors give thanks to Quinten Vanhellemont and Kevin Ruddick for free ACOLITE contribution.

**Conflicts of Interest:** The authors declare no conflict of interest.

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
