# Peer review of "A Simple Empirical Band-Ratio Algorithm to Assess Suspended Particulate Matter from Remote Sensing over Coastal and Inland Waters of Vietnam: Application to the VNREDSat-1/NAOMI Sensor"

_water, doi:10.3390/w12092636_

Round 1
Reviewer 1 Report
A simple band-ratio algorithm to assess suspended particulate matter from remote sensing over coastal and inland waters of Vietnam: application to the VNREDSat-1/NAOMI sensor.
Vietnam recently (when?) launched a new remote sensing satellite that is being used to monitor water quality of coastal and inland waters of the country. The authors develop a multiband algorithm for suspended particulate matter (SPM) and validate it with in situ data in three regions of Vietnam. They then compare the algorithm to four algorithms previously developed for other regions. There was no direct matchup data (?) so an indirect test using Landsat data was also done.
Abstract
line 24 replace allowing with enabling
line 36 rather than comma use a decimal point
Line 37 replace allows to reduce with reduces
Line 39 replace estimate with estimating
Line 40 replace has been with was
Line 41 replace showing with yielding
Introduction
Clear statement of objective in developing an algorithm for suspended particulate matter
Table 1 and Figure 1 may be better placed in the methods section.
Line items
Line 46 replace “strong threatening” with “impact”
Line 47 delete “which”
Line 48 replace “a” with “the”
Line 53 Vietnamese
Line 58 chl-a and chlorophyll a are interchangeable
Line 61 remove “so-called”
Line 62 replace “matters” with “substances”
Line 66 delete “for their proper observations”
Line 68 replace “probably retained” with “has attracted”
Line 69 replace “subjects” with processes”
Line 72 replace “the top of atmosphere signal recorded by the sensor has been corrected from the atmospheric and interface signals” with “atmospheric correction”
Line 79 when was the satellite launched? (good to know when dataset starts)
Line 80 cite Table 1, Figure 1
Line 83 replace “purpose” with “objective”, “present” with “current”, “propose” with “develop”
Line 87 replace “methods” with “algorithms”
Line 90 don’t capitalize inland
Line 92 capitalize Peninsula
Material and Methods
Can you provide more information on the methodology and dates of the in situ dataset? It appears you glean this from several programs.
For these three datasets, the D-DS was used to develop the algorithm, and the V-DS was used to validate the algorithm, so I assume those in situ measurements have matching VNREDSat-1 imagery. So I am a little unclear in the use of the M-DS dataset that apparently does not have matching VNREDSat-1 imagery, so instead the algorithm was used on LANDSAT imagery after considering appropriate atmospheric correction? What is the utility of that step?
Line items
Line 96 delete contrasted
Line 99-100 capitalize River
Line 107 replace gather with comprise
Line 138 capitalize Estuary
Line 144 replace “considering” with “using”
Line 148 replace “in a first step” with “initially”
Line 149 replace “vary as well” with “also vary”
Results and discussion
The newly developed band ratio algorithm appeared to perform very well in a wide range of SPM values of Vietnamese waters. In the validation step, it was an improvement from four other SPM algorithms developed for other regions, although three out of the four performed adequately in Vietnamese waters. I was unclear why the secondary validation step with OLI images was needed if matches were available in first two steps (development and initial validation)? The poor performance of the algorithm for the Cuviet Estuary appears to be due to the narrow range of SPM values. Good use of the flooding event to illustrate use of algorithm.
Line 194 replace “can be” with “was”
Line 195 delete “relationship”, replace “feature” with “scatter”
Line 196 replace “illumination condition” with “light regime”?
Line 197 delete “formalisms” just use “band ratio”
Line 203 replace “the previous results” with “this finding”
Line 211 each of the sensors
Line 213 replace “considering” with “in”
Line 219 delete “more”
Line 241 delete “an”
Line 243 capitalize Estuary
Figure 8 x-axis measured
Line 285 replace besides with additionally
Line 287 replace “deeper researches” with “further research”
Line 292 a large flooding event
Line 296 replace fast with rapid
Line 298 river plume area, replace into with in
Author Response
Response to Reviewer 1 Comments
Point 1: Vietnam recently (when?) launched a new remote sensing satellite that is being used to monitor water quality of coastal and inland waters of the country. The authors develop a multiband algorithm for suspended particulate matter (SPM) and validate it with in situ data in three regions of Vietnam. They then compare the algorithm to four algorithms previously developed for other regions. There was no direct matchup data (?) so an indirect test using Landsat data was also done.
Response 1:
- VNREDSat-1 currently is the first and unique multi spectral optical satellite in Vietnam. It was launched on 7th May, 2013 (it is now specified in the introduction) but this is the first time used for water quality monitoring over coastal and inland waters of Vietnam. The date is now specified.
- Unlucky, as it is explained in the original manuscript, that we have not direct matchup data so we used Landsat-8 top of atmosphere data for testing our SPM algorithm.
Abstract
Point 2: line 24 replace allowing with enabling
Response 2: Done
Point 3: line 36 rather than comma use a decimal point
Response 3: Done
Point 4: Line 37 replace allows to reduce with reduces
Response 4: Done
Point 5: Line 39 replace estimate with estimating
Response 5: Done
Point 6: Line 40 replace has been with was
Response 6: Done
Point 7: Line 41 replace showing with yielding
Response 7: Done
Introduction
Point 8: Clear statement of objective in developing an algorithm for suspended particulate matter
Response 8: The first Vietnamese optical satellite, VNREDSat-1, carries the New AstroSat Optical Modular Instrument (NAOMI) which has 4 multi-spectral bands (Table 1) at 10 m of spatial resolution and 1 panchromatic band at 2.5 m [23, 24]. The water pixel extraction and atmospheric correction algorithms for VNREDSat-1 have recently been developed (Ngoc et al, 2019), allowing to estimate Rrs(λ) at each NAOMI spectral band. This successful work opens the gate to assess biogeochemical parameters (SPM, Chlorophyll-a, etc.) from this sensor. Due to the important information carried by SPM concentration for better understanding regional coastal and inland water dynamics, the present paper focus on the development of a NAOMI SPM inversion algorithm over Vietnam coastal and inland waters (hereafter referred to as V1SPM).
Point 9: Table 1 and Figure 1 may be better placed in the methods section.
Response 9: Figure 1, which describes the Spectral Response Functions for each VNREDSat-1 band, used to process the in situ hyper-spectral Rrs data was already in the Method section in the original manuscript. Regarding to Table 1, which provides general information on the Naomi sensor, we believe that it should be placed in the introduction where VNREDSat-1 is first introduced.
Line items
Point 10: Line 46 replace “strong threatening” with “impact”
Response 10: Done
Point 11: Line 47 delete “which”
Response 11: Done
Point 12: Line 48 replace “a” with “the”
Response 12: Done
Point 13: Line 53 Vietnamese
Response 13: Done
Point 14: Line 58 chl-a and chlorophyll a are interchangeable
Response 14: We propose to keep it as its acronyms
Point 15: Line 61 remove “so-called”
Response 15: Done
Point 16: Line 62 replace “matters” with “substances”
Response 16: Done
Point 17: Line 66 delete “for their proper observations”
Response 17: Done
Point 18: Line 68 replace “probably retained” with “has attracted”
Response 18: Done
Point 19: Line 69 replace “subjects” with processes”
Response 19: Done
Point 20: Line 72 replace “the top of atmosphere signal recorded by the sensor has been corrected from the atmospheric and interface signals” with “atmospheric correction”
Response 20: Done
Point 21: Line 79 when was the satellite launched? (good to know when dataset starts)
Response 21: This is now specified in the introduction.
Point 22: Line 80 cite Table 1, Figure 1
Response 22: Done
Point 23: Line 83 replace “purpose” with “objective”, “present” with “current”, “propose” with “develop”
Response 23: Done
Point 24: Line 87 replace “methods” with “algorithms”
Response 24: Done
Point 25: Line 90 don’t capitalize inland
Response 25: Done
Point 26: Line 92 capitalize Peninsula
Response 26: Done
Material and Methods
Point 27: Can you provide more information on the methodology and dates of the in situ dataset? It appears you glean this from several programs.
For these three datasets, the D-DS was used to develop the algorithm, and the V-DS was used to validate the algorithm, so I assume those in situ measurements have matching VNREDSat-1 imagery. So I am a little unclear in the use of the M-DS dataset that apparently does not have matching VNREDSat-1 imagery, so instead the algorithm was used on LANDSAT imagery after considering appropriate atmospheric correction? What is the utility of that step?
Response 27: The in situ dataset (including TriOS hyperspectral data: remote sensing reflectances from 400nm to 900nm with 1nm resolution; and SPM value by sampling analysis) gathered 205 points which are represented in Figure 2. None of these in situ data was unfortunately matching with VNREDSat-1 images, avoiding the possibility to perform usual mathcup exercise in order to validate the satellite output taking into account all the uncertainties along the processing chain from the TOA signal. In this context, an indirect validation was performed using OLI data for which matchups are existing. In practice, OLI data were processedusing the VNREDSat-1 processing chain and validated with in situ measurements, while in a second step, these validated OLI data were compared with VNREDSat-1 outputs.
This is now better specified in the introduction:
“Because, unfortunately, no in situ SPM measurements have been collected during VNREDSat overpass, this new SPM algorithm is applied to Landsat-8/OLI observations performed nearly simultaneously to some water sampling in Vietnam and Cambodia coastal and inland waters. For that purpose, the OLI scenes have been processed using the NAOMI algorithms, that is the NAOMI RED-NIR atmospheric correction [22] and V1SPM algorithms, to first assess the coherence of NAOMI algorithms to assess SPM. The pertinence of this match-up exercise procedure, developed from OLI observations but processed using the NAOMI algorithms, has already been partly discussed in [22] for the validation of the Red-NIR atmospheric correction algorithm. Inter-comparison of the NAOMI SPM products is then performed at the Camau Peninsula (South of Vietnam) observed nearly simultaneously by NAOMI, processed using the Red-NIR and V1SPM algorithms, and OLI, processed using the last version of ACOLITE (Vanhellemont, 2019). “
Line items
Point 28: Line 96 delete contrasted
Response 28: Done
Point 29: Line 99-100 capitalize River
Response 29: Done
Point 30: Line 107 replace gather with comprise
Response 30: Done
Point 31: Line 138 capitalize Estuary
Response 31: Done
Point 32: Line 144 replace “considering” with “using”
Response 32: Done
Point 33: Line 148 replace “in a first step” with “initially”
Response 33: Done
Point 34: Line 149 replace “vary as well” with “also vary”
Response 34: Done
Results and discussion
Point 35: The newly developed band ratio algorithm appeared to perform very well in a wide range of SPM values of Vietnamese waters. In the validation step, it was an improvement from four other SPM algorithms developed for other regions, although three out of the four performed adequately in Vietnamese waters. I was unclear why the secondary validation step with OLI images was needed if matches were available in first two steps (development and initial validation)? The poor performance of the algorithm for the Cuviet Estuary appears to be due to the narrow range of SPM values. Good use of the flooding event to illustrate use of algorithm.
Response 35: As mentioned in a previous Response, due to the lack of matchup data (concomitant in situ and satellite measurements) for VNREDSat-1 the satellite validation was here performed indirectly based on OLI data.
Point 36: Line 194 replace “can be” with “was”
Response 36: Done
Point 37: Line 195 delete “relationship”, replace “feature” with “scatter”
Response 37: Done
Point 38: Line 196 replace “illumination condition” with “light regime”?
Response 38: We propose to keep it
Point 39: Line 197 delete “formalisms” just use “band ratio”
Response 39: Done
Point 40: Line 203 replace “the previous results” with “this finding”
Response 40: We propose to keep it
Point 41: Line 211 each of the sensors
Response 41: Done
Point 42: Line 213 replace “considering” with “in”
Response 42: Done
Point 43: Line 219 delete “more”
Response 43: Done
Point 44: Line 241 delete “an”
Response 44: Done
Point 45: Line 243 capitalize Estuary
Response 45: Done
Point 46: Figure 8 x-axis measured
Response 46: Done
Point 47: Line 285 replace besides with additionally
Response 47: Done
Point 48: Line 287 replace “deeper researches” with “further research”
Response 48: Done
Point 49: Line 292 a large flooding event
Response 49: Done
Point 50: Line 296 replace fast with rapid
Response 50: Done
Point 51: Line 298 river plume area, replace into with in
Response 51: Done
Reviewer 2 Report
Dear Editor and Authors, the paper flows and visualization is of good quality. I don't have any particular requirements and sauggestions to improve the paper, apart the Conclusions section.
Conclusions must resume the main findigns, and it is very unusual to see a Figure in this section. I suggest to group comments, observations and future works in a separate section Discussion, with Fig.10, and just mentioning the main findings of this work in Conclusions.
Author Response
Response to Reviewer 2 Comments
Dear Editor and Authors, the paper flows and visualization is of good quality. I don't have any particular requirements and sauggestions to improve the paper, apart the Conclusions section.
Thank you very much for your kindly comments
Point 1: Conclusions must resume the main findigns, and it is very unusual to see a Figure in this section. I suggest to group comments, observations and future works in a separate section Discussion, with Fig.10, and just mentioning the main findings of this work in Conclusions.
Response 1: We took into account the reviewer’s recommendation, removing the summary previously documented in the conclusion further displacing the Figure 10 and corresponding text to a specific section of the results and discussion (3.4. Illustration of the monitoring of a flooding event from VNREDSat-1).
Reviewer 3 Report
We have here a well-presented study of a regional, empirical algorithm for retrieving SPM distribution in the coastal waters of Vietnam. I see some error sources that are not treated in the manuscript that I would like to see discussed by the authors. The graphical presentations of the analysis of the efficacy of the proposed algorithm, VISPM, need to be improved. I will write my comments on the page where I wish to point out what is needed.
Page 1, line 2. Replace “A simple band-ratio algorithm ...” with A simple empirical band-ratio algorithm ...”
Page 3, lines 87-89. Since this paper represents a validation of the VISPM algorithm, I get a little nervous that it is not a match-up between data from the VNREDSat-1/NAOMI sensor and contemporaneously collected SPM concentrations. Perhaps a few words are needed about how well the derivation of Landsat-8/OLI observations, in-situ hyperspectral observations, and VNREDSat-/NAOMI compare during contemporaneous observations of all three together.
Page 3, lines 103-105. In these lines it is pointed out that a significant portion of the SPM data can be considered “relatively low.” Concentration data on the low end have a significant problem, salt retention error (Stavn et al. 2009; Röttgers et al. 2014). At the lowest concentration the salt retention mass is the same order as the SPM mass. The salt retention error is significant up to an SPM concentration of about 7 g.m-3. This is problematic even at somewhat higher concentrations if a mapping and a gradient of concentrations is desired and Figure 3 establishes that the majority of SPM concentrations used to validate the algorithm proposed here is in this concentration region.
Page 5, lines 130-134. The geometric means published for the algorithm validation and the
Landsat-8/OLI dataset are well within the range established as problematic due to salt retention error.
Page 6, lines148-150. The authors mention the difficulty of comparing remote-sensing algorithms for SPM derived for different regions. This comes about from all work with SPM; it is an unknown mixture of suspended organics and inorganics, with radically different refractive indices, which constitutes an unknown systematic error in all such algorithms.
Page 7, lines187-192. The authors compare single-band and band-ratio methods for remote estimations of SPM and represent this with a log-log plot. My response to log-log plots is usually skeptical of the point an author is trying to make but I believe the authors have demonstrated here that their log-log plots are useful for demonstrating overall trends. The point being made is that the band-ratio plot is more uniform that the single band plot with the exception of the lower concentrations of SPM. Refer back to salt-retention comments of this point. And, the polynomial curve for the single-band algorithm in Figure 5(a) could easily be broken into two linear plots diverging at about 5 g.m-3. A possible explantion for the divergence of the two curves is that below 5g.m-3 the SPM might be predominantly organic and above this value the SPM might be predominantly inorganic.
Page 8, lines 214-217. It is pointed out here that the Doxaran et al. algorithm had the worst performance of the algorithms considered for comparison. The Doxaran et al. algorithm was derived from the Gironde estuary in France where it was stated that organic matter in the SPM samples collected was at less than 1.8% concentration. Thus, an assumption of SPM composed of essentially mineral matter was made by those authors. This is the only SPM algorithm study mentioned in thus paper that is relatively free of the systematic error in all presumptive SPM algorithms. Thus their algorithm was based on a band ratio of SPOT-HRV bands XS3(790-890 nm)/XS2(610-680 nm). This band-ratio algorithm worked well in the Gironde because the authors had established the suspended mineral nature of the SPM and the NIR XS3 band is most sensitive to the concentration of mineral matter. Figure 6(a) shows overestimation of SPM by the Doxaran et al. Model because the SPM in this study is an unknown quantity of organic vs mineral and the XS3/Xs2 band is greatly magnifying the mineral effects. However, at the higher concentrations it might be presumed that the SPM on the Vietnamese coast is predominantly mineral in nature and the Doxaran et al. algorithm was able to yield plausible estimates.
Page 8, lines 224-226. This section needs to be expanded to include a more detailed description of the nature of the algorithms investigated for the Vietnamese coast. When using the regression line in comparison with the 1:1 line, the Nechad algorithm does indeed underestimate SPM at levels greater than 100 g.m-3 but it seriously overestimates it below this concentration. The next two algorithms, Siswanto and Han-OLI tend to underestimate SPM for all concentrations. Indeed the VISPM algorithm has the highest precision but it also has a tendency to underestimate SPM above about 100 g.m-3 while there is a tendency to overestimate below that concentration. And the overestimation tendency comes at low concentration which are subject to salt-retention error.
Page 9, lines 226-227. The plots presented here are not correct. The purpose of the graphs presented in this section is to investigate the efficiency of the various algorithms to predict SPM concentration. Therefore the estimated SPM concentration is the independent variable plotted on the x-axis and the measured SPM is the dependent variable plotted on the y-axis. The plotting at present is demonstrating the efficiency of the measured SPM concentration to predict the algorithmic SPM concentration.
Page 12, lines 248-255. Figure 8 is also plotted incorrectly.
Page 14, lines 287-289. An attempt to explore the variability in an SPM algorithm will never be as successful as exploring this in a Chla algorithm because Chla is a definable molecular entity with a predictable refractive index while SPM is not.
References:
Röttgers, R., Heymann, K., and H. Krasemann. 2014. Suspended matter concentrations in coastal waters: Methodological improvements to quantify individual measurement uncertainty. Estuar., Coast. Shelf Sci., 151: 148-l55.
Stavn, R.H., Rick, H.J., Falster, A.V., 2009. Correcting the errors from variable sea salt retention and water of hydration in loss on ignition analysis: implications for studies of estuarine and coastal waters. Estuar., Coast. Shelf Sci., 81: 575-582.
Author Response
Response to Reviewer 3 Comments
We have here a well-presented study of a regional, empirical algorithm for retrieving SPM distribution in the coastal waters of Vietnam. I see some error sources that are not treated in the manuscript that I would like to see discussed by the authors. The graphical presentations of the analysis of the efficacy of the proposed algorithm, VISPM, need to be improved. I will write my comments on the page where I wish to point out what is needed.
Point 1: Page 1, line 2. Replace “A simple band-ratio algorithm ...” with A simple empirical band-ratio algorithm ...”
Response 1: done
Point 2: Page 3, lines 87-89. Since this paper represents a validation of the VISPM algorithm, I get a little nervous that it is not a match-up between data from the VNREDSat-1/NAOMI sensor and contemporaneously collected SPM concentrations. Perhaps a few words are needed about how well the derivation of Landsat-8/OLI observations, in-situ hyperspectral observations, and VNREDSat-/NAOMI compare during contemporaneous observations of all three together.
Response 2: We fully agree with the reviewer on this point. This is now better explained in the introduction where the following sentences have been added: “Because, unfortunately, no in situ SPM measurements have been collected during VNREDSat overpass, this new SPM algorithm is applied to Landsat-8/OLI observations performed nearly simultaneously to some water sampling in Vietnam and Cambodia coastal and inland waters. For that purpose, the OLI scenes have been processed using the NAOMI algorithms, that is the NAOMI RED-NIR atmospheric correction [22] and V1SPM algorithms, to first assess the coherence of NAOMI algorithms to assess SPM. The pertinence of this match-up exercise procedure, developed from OLI observations but processed using the NAOMI algorithms, has already been partly discussed in [22] for the validation of the Red-NIR atmospheric correction algorithm. Inter-comparison of the NAOMI SPM products is then performed at the Camau Peninsula (South of Vietnam) observed nearly simultaneously by NAOMI, processed using the Red-NIR and V1SPM algorithms, and OLI, processed using the last version of ACOLITE (Vanhellemont, 2019). “
Concerning the use of in situ hyperspectral measurements, we already described in the original manuscript that both OLI and VNREDSat-1 bands have been recomputed using the spectral response function of each sensor as already done In Ngoc et al. (2019).
Point 3: Page 3, lines 103-105. In these lines it is pointed out that a significant portion of the SPM data can be considered “relatively low.” Concentration data on the low end have a significant problem, salt retention error (Stavn et al. 2009; Röttgers et al. 2014). At the lowest concentration the salt retention mass is the same order as the SPM mass. The salt retention error is significant up to an SPM concentration of about 7 g.m-3. This is problematic even at somewhat higher concentrations if a mapping and a gradient of concentrations is desired and Figure 3 establishes that the majority of SPM concentrations used to validate the algorithm proposed here is in this concentration region.
Response 3: We agree with the reviewer on that point. However, the impact of Salt on the SPM measurements has been considered during the measurement procedure, which tends to limit this impact. This is now well explained in the method section where the following sentences have been added: “The protocol used to calculate the Rrs spectra from radiometric measurements is fully described in [9]. Due to the large fluctuation of salinity in the data set (28.24 ± 0.14), driven by impact of fresh water inputs at some stations, and which may bias the SPM measurements (Stavn et al., 2009; Röttgers et al., 2014), a great care has been taken to properly remove salt from the filters which have gently been washed with deionized water. Three replicates have been made at each station, and only stations with standard deviation lower than 20% have been kept allowing to reduce the impact of measurements uncertainties.”
Other information, regarding to the protocol used to assess Rrs from radiometric measurements has also been added in the method section.
Point 4: Page 5, lines 130-134. The geometric means published for the algorithm validation and the Landsat-8/OLI dataset are well within the range established as problematic due to salt retention error.
Response 4: Salt retention has been considered as much as possible, as described above.
Point 5: Page 6, lines148-150. The authors mention the difficulty of comparing remote-sensing algorithms for SPM derived for different regions. This comes about from all work with SPM; it is an unknown mixture of suspended organics and inorganics, with radically different refractive indices, which constitutes an unknown systematic error in all such algorithms.
Response 5: We fully agree with the comment, which is a basic problem already raised in many other papers. Initially, four existing algorithms were considered. The selected SPM inversion algorithms differ in terms of the sensor specific development assumption and also vary regarding the SPM range over which each algorithm has been developed, as well as from the development data set used which may different in terms of inherent optical properties driven by the concentration, chemical nature (mineral vs. organic), and particulate size distribution of the optically significant matter.
Point 6: Page 7, lines187-192. The authors compare single-band and band-ratio methods for remote estimations of SPM and represent this with a log-log plot. My response to log-log plots is usually skeptical of the point an author is trying to make but I believe the authors have demonstrated here that their log-log plots are useful for demonstrating overall trends. The point being made is that the band-ratio plot is more uniform that the single band plot with the exception of the lower concentrations of SPM. Refer back to salt-retention comments of this point. And, the polynomial curve for the single-band algorithm in Figure 5(a) could easily be broken into two linear plots diverging at about 5 g.m-3. A possible explantion for the divergence of the two curves is that below 5g.m-3 the SPM might be predominantly organic and above this value the SPM might be predominantly inorganic.
Response 6: Using the bbp/bp ratios, a good driver of the organic vs. mineral part of the suspended particulate matter, available for each data points, we tried to find an explanation to the observed behavior. However, no significant pattern has been observed in terms of organic vs. mineral.
Point 7: Page 8, lines 214-217. It is pointed out here that the Doxaran et al. algorithm had the worst performance of the algorithms considered for comparison. The Doxaran et al. algorithm was derived from the Gironde estuary in France where it was stated that organic matter in the SPM samples collected was at less than 1.8% concentration. Thus, an assumption of SPM composed of essentially mineral matter was made by those authors. This is the only SPM algorithm study mentioned in thus paper that is relatively free of the systematic error in all presumptive SPM algorithms. Thus their algorithm was based on a band ratio of SPOT-HRV bands XS3(790-890 nm)/XS2(610-680 nm). This band-ratio algorithm worked well in the Gironde because the authors had established the suspended mineral nature of the SPM and the NIR XS3 band is most sensitive to the concentration of mineral matter. Figure 6(a) shows overestimation of SPM by the Doxaran et al. Model because the SPM in this study is an unknown quantity of organic vs mineral and the XS3/Xs2 band is greatly magnifying the mineral effects. However, at the higher concentrations it might be presumed that the SPM on the Vietnamese coast is predominantly mineral in nature and the Doxaran et al. algorithm was able to yield plausible estimates.
Response 7: We fully agree with that comment, and the following sentence has been added in the new version of the paper: “This result was expected as the empirical SPOT-5-Doxaran algorithm was developed from 42 in situ SPM data points with concentrations ranging from 35 to 2072 mg.l-1.”
Point 8: Page 8, lines 224-226. This section needs to be expanded to include a more detailed description of the nature of the algorithms investigated for the Vietnamese coast. When using the regression line in comparison with the 1:1 line, the Nechad algorithm does indeed underestimate SPM at levels greater than 100 g.m-3 but it seriously overestimates it below this concentration. The next two algorithms, Siswanto and Han-OLI tend to underestimate SPM for all concentrations. Indeed the VISPM algorithm has the highest precision but it also has a tendency to underestimate SPM above about 100 g.m-3 while there is a tendency to overestimate below that concentration. And the overestimation tendency comes at low concentration which are subject to salt-retention error.
Response 8: This section has been better specified has follows:
“A gain in precision provided by the V1SPM-NAOMI model when compared to Nechad-OLI, Siswanto and Han-OLI. Indeed, Nechad-OLI (Figure 6b) estimates are sharply underestimated at SPM levels higher than 100 g.m-3 while an overestimation is found for SPM loads lower than the latter concentration. Siswanto and Han-OLI (Figures 6c and 6, respectively) are conversely underestimating SPM over the whole range of values in V-DS dataset. Although the general better performance for the V1SPM algorithm (Figure 6e) , a slight under-estimation of the SPM loads is observed at the highest levels (> 100 g.m-3), while a low SPM, where measurement uncertainties can occur (e.g. salt retention issues) are slightly overestimated.”
is in particularly observed at higher SPM levels (> 100 g.m-3) where Nechad-OLI, Siswanto and Han-OLI tend to underestimate the SPM concentration (Figure 6e).
Point 9: Page 9, lines 226-227. The plots presented here are not correct. The purpose of the graphs presented in this section is to investigate the efficiency of the various algorithms to predict SPM concentration. Therefore, the estimated SPM concentration is the independent variable plotted on the x-axis and the measured SPM is the dependent variable plotted on the y-axis. The plotting at present is demonstrating the efficiency of the measured SPM concentration to predict the algorithmic SPM concentration.
Response 9: We do not agree with the reviewer. The way here chosen is, according to our knowledge, the usual way considered in similar plots. Further, it allows an intuitive understanding of the results with a direct view of the under(over) estimated values considering the location of the points with respect to the 1:1 line.
Point 10: Page 12, lines 248-255. Figure 8 is also plotted incorrectly.
Response 10: please refer to the previous answer.
Point 11: Page 14, lines 287-289. An attempt to explore the variability in an SPM algorithm will never be as successful as exploring this in a Chla algorithm because Chla is a definable molecular entity with a predictable refractive index while SPM is not.
Response 11: We agree with the reviewer. However, we also believe that the bio-optical environment (dissolved vs. particulate matter, organic vs. inorganic bulk particulate matter, particle size distribution, etc) has also a great importance (as it impact IOPs and then Rrs differently) on the dispersion found in the satellite products retrieval in addition to the factors driving the inner natural variability of the aimed product.
Reviewer 4 Report
This study tried to develop a SPM estimation algorithm for the VNREDSat-1/NAOMI sensor, especially in the water region of Vietnam. In particular, ratio of band 3 and band 2 was found to be the most accurate representation of SPM. This study also compared this algorithm with other existing three and found the proposed one was the best.
Overall, this study has good implications for water environment monitoring and application in Vietnam based on VNREDSat-1/NAOMI sensor. However, there are some concerns that should be addressed before publication. Please refer to the following comments for the details.
- P3 L88-89: Due to the lack of in-situ SPM measurements during VNREDSsat overpass, this study used Landsat-8/OLI data for representing the VNREDSsat data. This is the trickiest problem for the whole study. The spectral, spatial, and internal physical characteristics of them are not the same. I strongly suggest that the authors proved the feasibility and rationality of this replacement.
- P5 L128-134: Two tables and one figure (actually two if include Figure 2) have been used for describing the in-situ measurements. I am thinking of merging them for a concise expression. In addition, the most important parameter, measurement time of these data has not been provided.
- P13 L276: Please add the explanation of Figure 9(e).
- P14 L299: I suggest move this initial application to the result section.
- Seasonal variation of this algorithm should also be discussed.
Author Response
Response to Reviewer 4 Comments
This study tried to develop a SPM estimation algorithm for the VNREDSat-1/NAOMI sensor, especially in the water region of Vietnam. In particular, ratio of band 3 and band 2 was found to be the most accurate representation of SPM. This study also compared this algorithm with other existing three and found the proposed one was the best.
Overall, this study has good implications for water environment monitoring and application in Vietnam based on VNREDSat-1/NAOMI sensor. However, there are some concerns that should be addressed before publication. Please refer to the following comments for the details.
Point 1: P3 L88-89: Due to the lack of in-situ SPM measurements during VNREDSsat overpass, this study used Landsat-8/OLI data for representing the VNREDSsat data. This is the trickiest problem for the whole study. The spectral, spatial, and internal physical characteristics of them are not the same. I strongly suggest that the authors proved the feasibility and rationality of this replacement.
Response 1: The in situ dataset (including TriOS hyperspectral data: remote sensing reflectances from 400nm to 900nm with 1nm resolution; and SPM value by sampling analysis) gathered 205 points which are represented in Figure 2. None of these in situ data was unfortunately matching with VNREDSat-1 images, avoiding the possibility to perform usual mathcup exercise in order to validate the satellite output taking into account all the uncertainties along the processing chain from the TOA signal. In this context, an indirect validation was performed using OLI data for which matchups are existing. In practice, OLI data were processed using the VNREDSat-1 processing chain and validated with in situ measurements, while in a second step, these validated OLI data were compared with VNREDSat-1 outputs.
It is now better specified in the introduction
“Because, unfortunately, no in situ SPM measurements have been collected during VNREDSat overpass, this new SPM algorithm is applied to Landsat-8/OLI observations performed nearly simultaneously to some water sampling in Vietnam and Cambodia coastal and inland waters. For that purpose, the OLI scenes have been processed using the NAOMI algorithms, that is the NAOMI RED-NIR atmospheric correction [22] and V1SPM algorithms, to first assess the coherence of NAOMI algorithms to assess SPM. The pertinence of this mMatch-up exercise procedure, developed from OLI observations but processed using the NAOMI algorithms, has already been partly discussed in [22] for the validation of the Red-NIR atmospheric correction algorithm. Inter-comparison of the NAOMI SPM products is then performed at the Camau Peninsula (South of Vietnam) observed nearly simultaneously by NAOMI, processed using the Red-NIR and V1SPM algorithms, and OLI, processed using the last version of ACOLITE (Vanhellemont, 2019). “
Point 2: P5 L128-134: Two tables and one figure (actually two if include Figure 2) have been used for describing the in-situ measurements. I am thinking of merging them for a concise expression. In addition, the most important parameter, measurement time of these data has not been provided.
Response 2: We do believe that the two tables should be kept as in the original version of the manuscript.While both Tables are indeed dealing with in situ data (Table 2 development and validation data sets and Table 3 matchup data set), a more precise information is provided for the mathcup data which information is not useful for the other 2 data sets. While the time of measurement is indeed not indicated in Table 3, a precision on the temporal criterion considered for selecting matchup data is now provided in the text as follows: “The time difference between in situ and satellite over-pass is always lower than 4 hours. “
Point 3: P13 L276: Please add the explanation of Figure 9(e).
Response 3: The Figure 9 caption has been modified including the panel (e)
Point 4: P14 L299: I suggest move this initial application to the result section.
Response 4: This has been performed creating a new results and discussion subsection: 3.4. Illustration of the monitoring of a flooding event from VNREDSat-1
Point 5: Seasonal variation of this algorithm should also be discussed.
Response 5: The characterization of the seasonal variation in the SPM loads over the investigated areas is indeed of high interest but is beyond the objective of the current paper, which was more likely dedicated to the algorjthm development/validation and illustration of the potential of the delivered information. However, a more detailed study of the temporal dynamics of SPM over Vietnamese coastal/inland waters using VNREDSat-1 will be further developed in the scope of another paper.
Round 2
Reviewer 4 Report
The authors have successfully addressed all my concerns. This study is valuable for assessing the SPM by using the VNREDSat-1/NAOMI sensor. Given NAOMI sensor was replaced by OLI sensor in this study, I looking forward more direct results conducted by in-situ data and NAOMI datasets in the near future. I recommend its publication is WATER.